# 👻 GHOST: Geometry-Guided Hallucination of Opaque Surface Textures

**Langxu Zhao** [* 1]   **Zuan Gu** [* 1]   **Tianhan Gao** [1]

## Abstract

Transparent objects pose a fundamental challenge for depth estimation and 3D reconstruction due to their violation of Lambertian assumptions, leading to severe geometry degradation in downstream tasks. To address this, we propose a novel geometry-guided preprocessing framework **GHOST** that leverages visual foundation models to transform transparent regions into opaque, structurally consistent representations without requiring downstream model retraining. Specifically, our pipeline utilizes (1) **TransDINO** and (2) **TransDecomp** to disentangle masks and transparency physical properties, while (3) **DAF-Net** recovers surface normal priors to encode geometric curvature. Subsequently, (4) **GeoSem-TransNet** integrates these multi-modal cues to synthesize a texture-rich opaque RGB image that preserves the transparent object's 3D structure. Extensive experiments demonstrate that our method significantly enhances the accuracy of state-of-the-art depth estimation and reconstruction models on transparent objects by restoring essential photometric cues.

## 1. Introduction

Mainstream depth estimation and 3D reconstruction models fundamentally assume Lambertian diffuse reflection (Kutulakos & Steger, 2008; Sajjan et al., 2020). However, transparent surfaces violate this via light transmission and specular reflection, causing severe geometric distortion. Recent academic efforts primarily pursue two directions: depth completion (Sajjan et al., 2020) and Neural Radiance Fields (NeRF) based reconstruction (Ichnowski et al., 2021).

[1]School of Software, Northeastern University, Shenyang, China. Correspondence to: Tianhan Gao <gaoth@mail.neu.edu.cn>.

*Proceedings of the 43rd International Conference on Machine Learning*, Seoul, South Korea. PMLR 306, 2026. Copyright 2026 by the author(s).

### 1.1. Existing Solutions and Limitations

Depth completion (Zhang & Funkhouser, 2018) uses deep networks to predict surface normals and boundaries, repairing depth via global optimization. However, these post-processing methods (Zhang et al., 2023) lack universality and efficiency due to their reliance on specific end-to-end training. Obtaining ground-truth depth for transparent objects is costly, requiring specialized hardware or powder spraying (Zhu et al., 2021). Consequently, these models generalize poorly to unseen materials or lighting. Meanwhile, NeRF-based approaches suffer from low inference efficiency, requiring multi-view inputs and lengthy per-scene training. This forces a trade-off: abandoning versatile SOTA models for narrow, task-specific ones—creating a "perception silo."

### 1.2. Universal Preprocessing Mechanism

Core challenges include expensive and scarce GT labels, low efficiency, and incompatibility with generic SOTAs. Visual Foundation Models (VFMs) (Awais et al.) offer a solution: self-supervised models like DINOv3 (Siméoni et al., 2025) capture dense features and structural boundaries of transparent objects despite textural variations. Since downstream models primarily support RGB, modifying only transparent pixels to mimic opaque counterparts can correct inference results. We propose GHOST (Geometry-guided Hierarchical Opaque Style Transfer), a universal pre-processor that "repaints" transparent regions into structurally consistent opaque objects before downstream processing. GHOST uses DINOv3 for region disentanglement, DAF-Net for geometric priors, and GeoSemTransNet for "opaquification." This resolves generalization issues without retraining downstream models. Our contributions include:

**TransDINO**: A U-Net-based segmentation model injecting DINOv3 features for robust transparent object mask extraction.

**TransDecomp**: A dual-head ViT architecture that disentangles physical properties by separately estimating transparency and foreground values.

**DAF-Net**: A surface normal estimator using DINOv3 deep features for structural integrity and shallow features for local

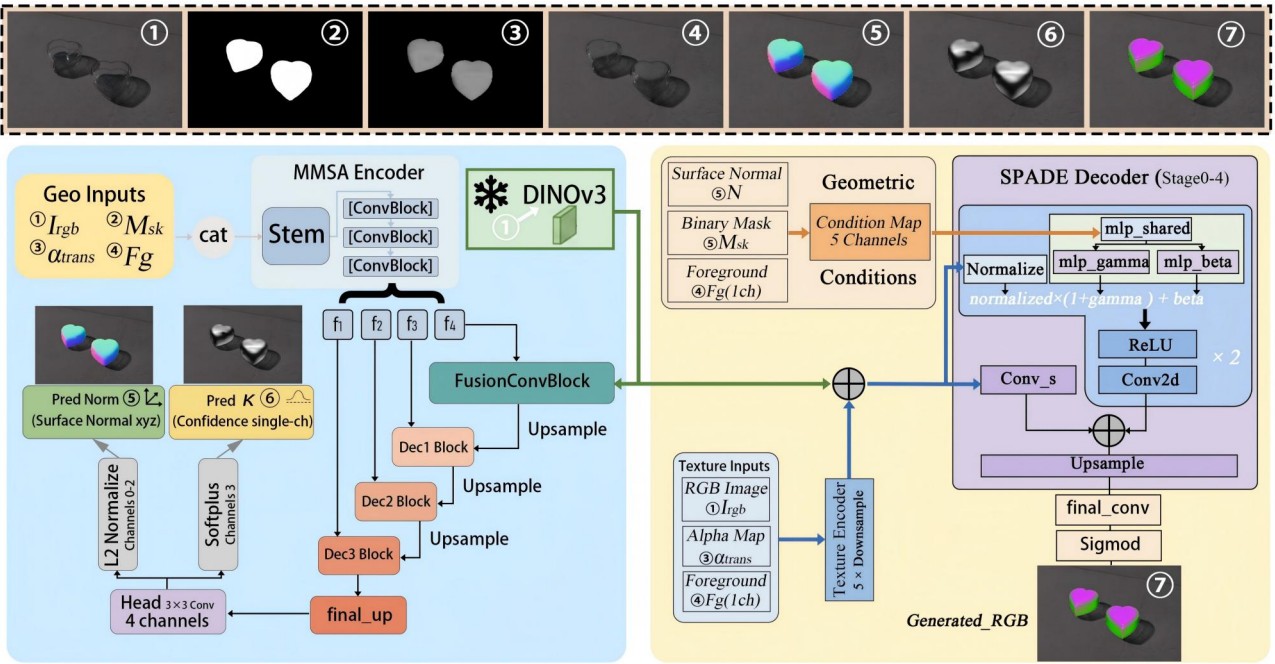

*Figure 1.* **Top**: (1)RGB image, (2)binary mask, (3)alpha matte, (4)foreground pixels × alpha, (5)surface normal map, (6)normal-confidence map, and (7)textured RGB image.**Bottom-Left**: In DAF-Net, the alpha matte and coarse foreground layer serve as geometric cues. An MMSA encoder processes the input, which is then fused with DINOv3 features and progressively decoded. Two dedicated heads finally produce the surface normal map and its confidence.**Bottom-Right**: Within GeoSemTransNet, the alpha matte and foreground image supply texture cues, while the normal map and foreground provide geometric conditions. A two-layer bilateral MLP in the SPADE module generates a shared activation map with channel-wise $\gamma$ and $\beta$. These are linearly combined with the normalized texture encoding, which—after fusion and upsampling—is projected to yield the final 3-channel texture image.

refraction and specular flow.

**GeoSemTransNet**: An encoder-decoder fusing DINOv3 features via SPADE (Park et al., 2019) modules to hierarchically inject geometric priors, ensuring geometry-guided texture synthesis.

## 2. Related Work

### 2.1. Depth Estimation & 3D Reconstruction

Recent 3D reconstruction has shifted from traditional geometry to deep learning. DUSt3R (Wang et al., 2024) regresses pointmaps end-to-end without calibration, yet fails on transparent objects as refraction violates its photometric consistency assumption. VGGT (Wang et al., 2025a) infers attributes via Camera Tokens but lacks transparency inductive biases; its DINO features often focus on background textures, causing "penetration" artifacts. While $\pi^3$ (Wang et al., 2025b) resolves view bias via permutation-equivariant architectures, its latent manifold learning lacks geometric interpretability for complex optics. MapAnything (Keetha et al., 2025) utilizes semantic priors, but these backfire on transparent objects due to a lack of surface-specific textures, leading to foreground-background confusion. Finally, DepthAnythingv3 (DA3) (Lin et al., 2025) achieves

SOTA via "Depth-Ray" representations but lacks physical modeling (e.g., refractive index). Without high-quality 3D transparent data, DA3 typically predicts background depth rather than the actual glass surface.

### 2.2. Transparent Object Segmentation

Transparent object perception is hindered by refraction and reflection, which couple object appearance with the background. Lacking internal textures, boundaries become primary cues. TransLab (Xie et al., 2020) uses edge branches for contouring but fails under low contrast. EBLNet (He et al., 2021) employs non-local attention for boundary refinement but ignores planar geometric constraints. For multi-material scenes, TROSNet (Sun et al., 2023b) uses multi-level fusion but suffers from reflection ambiguity due to heavy parameterization and lack of depth priors. Among multi-task and Transformer-based models, MODEST (Liu et al., 2025) and EGSA-PT (Omotara et al., 2025) fuse semantic and geometric features, though the former's channel attention suppresses information and the latter requires high edge precision. Trans2Seg (Xie et al., 2021) uses learnable prototypes for retrieval but struggles with occlusion and stacking. Trans4Trans (Zhang et al., 2022) employs a dual-head architecture for unified segmentation at the cost of high

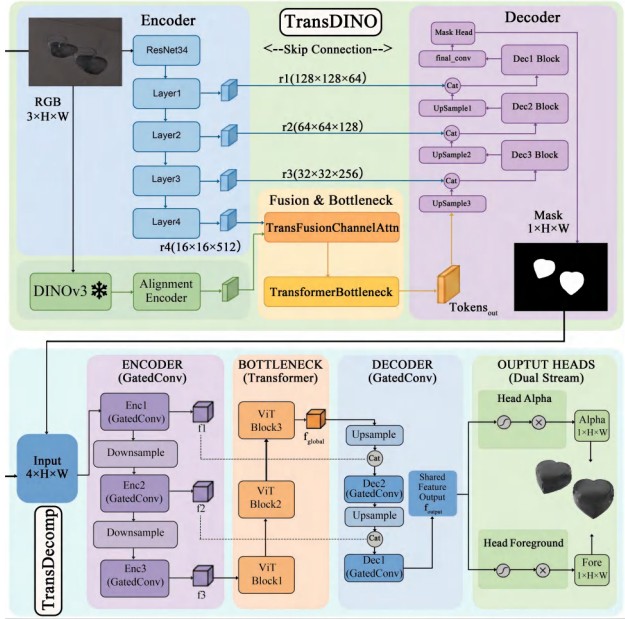

*Figure 2.* **Top**: TransDINO takes an RGB image as input and yields a high-quality semantic mask for transparent-object segmentation. **Bottom**: TransDecomp ingests RGB and mask and then regresses single-channel transparency and gray-scale foreground via dual heads. The result image: $RGB_{foreground(3ch)} \times alpha + RGB_{255,255,255} \times (1 - alpha)$ for visualization.

data dependency. Finally, TOSQ (Ma et al., 2025) treats segmentation as a dictionary look-up but remains vulnerable to cross-modal misalignment.

### 2.3. Depth Completion & NeRF for Transparent Objects

3D geometric perception is vital for robotic manipulation, yet commodity RGB-D sensors remain unreliable for transparent materials due to non-Lambertian light propagation(Sajjan et al., 2019). To address this, depth completion recovers corrupted maps, while NeRF provide continuous representations from multi-view data. **Depth Completion Models.** ClearGrasp (Sajjan et al., 2019) established a pipeline using surface normals and boundaries but requires complex post-processing and discards physical cues. HDCNet (Xie et al., 2025) achieves multi-scale completion by unifying Transformer, CNN, and Mamba (Gu & Dao, 2024) architectures. SwinDRNet (Dai et al., 2022) uses Swin Transformers (Liu et al., 2021) with confidence-based fusion, though it over-smooths textureless geometry. TranspareNet (Xu et al., 2021) leverages refractive offsets as priors, yet its efficiency depends on mask quality. ClueDepthGrasp (Hong et al., 2022) filters refractive noise for positional clues but struggles with multi-order refraction in complex shapes.

**NeRF-based Reconstruction Models.** NeRF-based methods capture view-dependent appearances. Dex-NeRF (Ich-

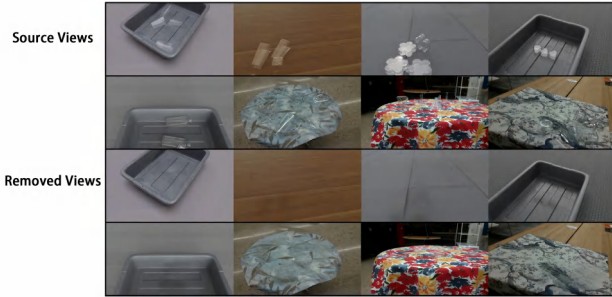

*Figure 3.* **Visual comparison of background generation**: The samples are selected from the ClearGrasp and ClearPose datasets. We utilize InpaintAnything to remove transparent objects, illustrating the contrast between the original RGB inputs and the cleaned background views.

nowski et al., 2021) identifies density gradients for grasping but requires hours of per-scene optimization. GraspNeRF (Dai et al., 2023) enables sparse-view inference via generalizable NeRF, yet demands $360°$ coverage and TSDF (Werner et al., 2014) supervision. NeRFrac (Zhan et al., 2023) models Snell's Law for consistency but remains sensitive to refractive index priors. Finally, NeRO (Liu et al., 2023) uses transmission gradients to reduce shape-radiance ambiguity, though its reliance on pre-trained SDFs slows the pipeline and fails on thin structures.

## 3. Method

We propose an image preprocessing framework specifically designed for transparent objects, comprising four topologically interconnected models that generate intermediate representations. The ultimate objective is to transform transparent regions into opaque surfaces that preserve the original structural details. The complete workflow is illustrated in the sequence of Fig.2 and Fig.1

### 3.1. Overview of the Proposed Pipeline

We first use TransDINO to extract segmentation masks $M_{sk}$. TransDecomp then predicts the foreground grayscale map $F_g$ and transparency $\alpha_{trans}$, which facilitate surface normal $N$ estimation via DAF-Net. Finally, GeoSemTransNet synthesizes structure-preserving opaque textures by using $I_{rgb}$, $F_g$, and $\alpha_{trans}$ as texture inputs, while being conditioned on $N$, $M_{sk}$, and $F_g$ for geometric consistency.

### 3.2. Transparent Object Localization via TransDINO

We propose TransDINO, a hybrid segmentation framework designed to synergize fine-grained spatial details from robust semantic priors from $DINOv3_{vits16}$. As illustrated in Fig.2, TransDINO adopts a U-Net architecture but introduces a novel dual-stream encoding mechanism and a Transformer-based Bottleneck to bridge the gap between low-level texture and high-level semantics.

*Table 1.* Quantitative comparison of TransDINO against state-of-the-art methods on ClearGrasp, ClearPose(Chen et al., 2022), Trans10K-v2(Xie et al., 2020), and TROS(Sun et al., 2023a) datasets.

| DATASET | METHOD | MIOU (%) |
|---|---|---|
| CLEARGRASP | EBLNET | 64.99 |
| | TROSNET (RGB-D$_{Input}$) | 71.40 |
| | TRANSDINO (OURS) | **92.44** |
| CLEARPOSE | MODEST | **90.98** |
| | EGSA-RGB (RGB$_{Edges}$) | 85.04 |
| | EGSA-RGB (DEPTH$_{Edges}$) | 87.30 |
| | TRANSDINO (OURS) | 85.21 |
| TRANS10KV2 | TRANSLAB | 69.00 |
| | TRANS2SEG | 72.15 |
| | TRANS4TRANS M | 75.14 |
| | TOSQ-256 | 77.47 |
| | TRANSDINO (OURS) | **86.82** |
| TROS | TRANSLAB | 50.72 |
| | TRANS4TRANS M | 39.22 |
| | EBLNET | 50.12 |
| | TROSNET (RGB-D$_{Input}$) | 57.23 |
| | TRANSDINO (OURS) | **57.94** |

**Dual-Source Feature Encoding**. To transcend RGB-only limitations, we employ a dual-branch encoder. A pre-trained ResNet-34 (He et al., 2015) backbone extracts multi-scale spatial features $\{F_{rgb}^i\}_{i=1}^4$, where the deepest map $F_{rgb}^4 \in \mathbb{R}^{C_u \times H \times W}$ provides rich local textures. Simultaneously, DINOv3 features are processed via a lightweight encoder to align with $F_{rgb}^4$, producing the semantic embedding $F_{sem}$.

**Attentive Feature Fusion.** TransDINO's core innovation is its progressive fusion strategy. We use a TransFusionChannelAttn module to align heterogeneous features, treating $F_{rgb}^4$ as Queries ($Q$) and $F_{sem}$ as Keys ($K$) and Values ($V$) within a spatial cross-attention (Vaswani et al., 2017) mechanism:

$$\tilde{F}_{cross} = \text{Softmax}\left(\frac{Q(F_{rgb}^4) \cdot K(F_{sem})^T}{\sqrt{d_k}}\right) V(F_{sem}) \quad (1)$$

This operation allows the local spatial features to query relevant global semantic contexts. To further suppress noise introduced by the cross-modal interaction, we employ a dual-pooling channel attention mechanism. Subsequently, we perform feature recalibration via the Hadamard product to emphasize informative channels:

$$F_{refine} = \tilde{F}_{cross} \odot M_c(\tilde{F}_{cross}) + F_{rgb}^4 \quad (2)$$

Here, the refined features are added back to the RGB stream $F_{rgb}^4$ as a residual connection. Finally, the refined stream is concatenated with the original auxiliary features $F_{sem}$ and fused through a convolutional block $\mathcal{H}$. To stabilize the training dynamics of this deep fusion module, we incorporate a learnable scaling factor. The final fused output $F_{out}$

is given by:

$$F_{out} = F_{concat} + \lambda \cdot \mathcal{H}(F_{concat}) \odot M_c(\mathcal{H}(F_{concat})) \quad (3)$$

where $F_{concat} = [F_{refine}, F_{sem}]$ and $\lambda$ is a learnable parameter initialized to 0. This initialization ensures better convergence by approximating an identity mapping during early training. The symmetric decoder upsamples bottleneck outputs, concatenating them with skip connections $\{F_{rgb}^i\}_{i=1}^4$. Stacked convolutions and batch normalization in decoder blocks progressively recover spatial resolution. Finally, a $1 \times 1$ convolution head generates the pixel-wise segmentation mask.

**Training Loss Function.** We employ a loss function that combines the Dice Loss ($L_{Dice}$) and the Intersection-over-Union Loss ($L_{IoU}$). The total objective function is formulated as a weighted sum:

$$L_{total} = \lambda_{Dice} L_{Dice} + \lambda_{IoU} L_{IoU} \quad (4)$$

where we set $\lambda_{Dice} = 0.5$ and $\lambda_{IoU} = 0.5$ to equally prioritize global geometric overlap and local pixel-wise accuracy during optimization.

### 3.3. Decomposing Alpha and Foreground Grayscale

We propose TransDecomp designed to decompose a transparent object region into $\alpha_{trans}$ and $F_g$. Under the 'weak refraction' assumption—where light displacement is small relative to the scale of thin-walled objects—approximating refraction as alpha blending provides a cost-effective computational solution. According to Snell's Law (Born & Wolf, 2013):

$$n_1 \sin(\theta_1) = n_2 \sin(\theta_2) \quad (5)$$

When the refractive index difference between medium 1 and medium 2 is minimal ($n_1 \approx n_2$), the incident angle $\theta_1$ and the refractive angle $\theta_2$ are nearly identical. Visually, light travels through the object in a nearly straight line with minimal bending or reflection. In such cases of weak refraction, the object appears highly transparent, and we can mathematically simplify the interaction as Alpha Blending. This allows us to ignore complex ray-tracing or geometric distortions while maintaining computational efficiency. In cases of strong refraction (where $n_2 \gg n_1$), the object surface appears less transparent or even opaque. While TransDecomp may struggle to extract ideal foreground pixels under these conditions, it compensates by generating a lower Alpha value, signaling to the pipeline that the surface is substantially non-transparent. This signal is vital for the subsequent GeoSemTransNet to prioritize surface-specific texture synthesis over background transmission.

As illustrated in Fig.2, our network processes $I_{rgb}$ concatenated with $M_{sk}$ through a U-Net structure enhanced with gated convolutions and a transformer bottleneck.

*Table 2.* Quantitative comparison of depth estimation and surface normal estimation. We compare the result of baseline models on original images versus images processed by GHOST framework. $\delta$ denotes threshold accuracy, while angular errors are measured in degrees.

| Method | Depth Metrics | | | | | | | | Normal Metrics | | | | | |
|---|---|---|---|---|---|---|---|---|---|---|---|---|---|---|
| | RMSE $\downarrow$ | REL $\downarrow$ | MAE $\downarrow$ | $\delta_{1.01} \uparrow$ | $\delta_{1.03} \uparrow$ | $\delta_{1.05} \uparrow$ | $\delta_{1.10} \uparrow$ | $\delta_{1.25} \uparrow$ | Mean $\downarrow$ | Median $\downarrow$ | RMSE $\downarrow$ | $11.25° \uparrow$ | $22.5° \uparrow$ | $30° \uparrow$ |
| DA3 | 0.023 | 0.031 | 0.017 | 0.298 | 0.647 | 0.805 | 0.944 | 0.997 | 32.30 | 28.44 | 38.07 | 15.91 | 42.03 | 56.26 |
| DA3 + GHOST | **0.018** | **0.023** | **0.013** | **0.388** | **0.766** | **0.890** | **0.972** | **0.998** | **21.26** | **17.52** | **26.23** | **30.26** | **64.95** | **78.33** |
| MoGe2 | 0.018 | 0.025 | 0.014 | 0.325 | 0.711 | 0.867 | 0.972 | **0.999** | 30.03 | 26.06 | 35.64 | 17.50 | 45.24 | 60.00 |
| MoGe2 + GHOST | **0.014** | **0.013** | **0.009** | **0.433** | **0.820** | **0.933** | **0.990** | **0.999** | **19.74** | **16.35** | **24.37** | **32.87** | **68.35** | **81.44** |
| DepthPro | 0.019 | 0.027 | 0.015 | 0.312 | 0.684 | 0.843 | 0.963 | 0.999 | 28.98 | 25.12 | 34.45 | 18.41 | 47.46 | 62.42 |
| DepthPro + GHOST | **0.018** | **0.024** | **0.013** | **0.351** | **0.731** | **0.872** | **0.972** | **0.999** | **24.98** | **21.45** | **29.84** | **21.35** | **54.04** | **69.88** |
| Metric3Dv2 | 0.028 | 0.038 | 0.021 | 0.211 | 0.540 | 0.734 | 0.926 | 0.995 | 42.91 | 40.86 | 48.52 | 7.40 | 23.73 | 35.69 |
| Metric3Dv2 + GHOST | **0.021** | **0.028** | **0.016** | **0.272** | **0.653** | **0.838** | **0.973** | **0.999** | **38.06** | **35.10** | **43.87** | **10.75** | **30.94** | **44.08** |
| VGGT | 0.021 | 0.028 | 0.015 | 0.334 | 0.701 | 0.846 | 0.955 | 0.996 | 31.24 | 27.06 | 37.30 | 18.99 | 45.59 | 58.80 |
| VGGT + GHOST | **0.020** | **0.022** | **0.012** | **0.427** | **0.787** | **0.904** | **0.977** | 0.996 | **20.20** | **16.60** | **25.09** | **33.62** | **67.46** | **79.98** |

*Figure 4.* **Top**: Qualitative results from various depth estimation models. We compare point clouds projected from raw input images versus GHOST-processed images, using predicted camera intrinsics. **Bottom**: Qualitative results from multiple 3D feed-forward reconstruction models, reconstructed from either raw image sequences or GHOST-processed image sequences.

**Gated Feature Encoding.** Unlike standard convolutions that treat all pixels equally, transparency estimation requires distinguishing between valid object features and background noise. We employ Gated Convolutions in the encoder, formulated as:

$$\mathbf{Y} = \phi(\mathbf{W}_f * \mathbf{X}) \odot \sigma(\mathbf{W}_g * \mathbf{X}) \qquad (6)$$

where $*$ denotes convolution, $\phi$ is the activation function ELU(Clevert et al., 2016), and $\sigma$ is the sigmoid function. This gating mechanism(Yu et al., 2019) allows the network to learn a dynamic feature, effectively suppressing irrelevant background textures during the encoding phase.

**Global Context Modeling via Transformer.** To resolve refractive ambiguities where local receptive fields fail, we insert a ViT bottleneck between the encoder and decoder. Multi-head self-attention layers process flattened feature maps to capture long-range dependencies, enabling the model to distinguish surface reflections from transmitted background patterns.

**Physics-Constrained Dual-Head Decoding.** The decoder utilizes symmetric gated convolutions with skip connections to recover spatial resolution. To enforce physical plausibility, we design two parallel prediction heads: 1. Opacity Head ($\mathcal{H}_\alpha$): Predicts the per-pixel alpha map $\hat{\boldsymbol{\alpha}} \in [0, 1]$. 2. Foreground Head ($\mathcal{H}_f$): Estimates the foreground intensity $\hat{\mathbf{F}} \in [0, 1]$, representing the object's specular reflections and intrinsic color effectively decoupled from the background. Crucially, the outputs are constrained by the known object mask $\mathbf{M}$ such that $\hat{\boldsymbol{\alpha}}_{i,j} = 0$ and $\hat{\mathbf{F}}_{i,j} = 0$ where $\mathbf{M}_{i,j} = 0$.

**Physical Reconstruction Formulation.** The background prior $\mathbf{B}$ (Fig. 3) is generated offline using InpaintAnything (Yu et al., 2023), serving as a pre-computed pseudo-label to guide the training process; by inputting $\mathbf{I}$ and $\mathbf{M}$, the model erases the transparent foreground and synthesizes occluded textures. Using this estimated $\mathbf{B}$, the reconstructed image $\hat{\mathbf{I}}$

*Table 3.* We compare generic depth estimation models with GHOST against specialized transparent object depth estimation methods (bottom section).

| Method | RMSE↓ | REL↓ | MAE↓ | $\delta_{1.05}$ ↑ | $\delta_{1.10}$ ↑ | $\delta_{1.25}$ ↑ |
|---|---|---|---|---|---|---|
| $DA3_{ghost}$ | 0.018 | 0.023 | 0.013 | 0.890 | 0.972 | 0.998 |
| $MoGe2_{ghost}$ | **0.014** | **0.013** | **0.009** | **0.933** | **0.990** | **1.000** |
| $DepthPro_{ghost}$ | 0.018 | 0.024 | 0.013 | 0.872 | 0.972 | 0.999 |
| $Metric3Dv2_{ghost}$ | 0.021 | 0.028 | 0.016 | 0.838 | 0.973 | 0.999 |
| $VGGT_{ghost}$ | 0.021 | 0.022 | 0.012 | 0.904 | 0.977 | 0.996 |
| HDCNet | 0.021 | 0.028 | 0.016 | 0.846 | 0.962 | 0.997 |
| SwinDRNet | 0.020 | 0.014 | **0.009** | 0.931 | 0.977 | 0.998 |
| TranspareNet | 0.028 | 0.029 | 0.013 | 0.753 | 0.903 | 0.992 |
| ClueDepthGrasp | 0.024 | 0.032 | 0.021 | 0.784 | 0.934 | 0.988 |
| DKT(Xu et al., 2025) | 0.021 | 0.025 | 0.013 | 0.897 | 0.979 | 0.997 |

*Table 4.* Quantitative comparison of 3D reconstruction quality. We scale Acc, Comp, CD up by 100 and scale ND down by 10.

| Method | Acc↓ | Comp↓ | CD↓ | ND↓ | F-Score↑ |
|---|---|---|---|---|---|
| DA3 | 1.17 | 3.28 | 2.23 | 6.90 | 45.91 |
| $DA3_{ghost}$ | **0.79** | **1.73** | **1.26** | **5.71** | **60.49** |
| DUSt3R | 0.57 | 1.63 | 1.10 | 8.37 | 77.36 |
| $DUSt3R_{ghost}$ | **0.25** | **0.77** | **0.51** | **7.01** | **88.66** |
| MapAnything | 1.76 | 1.58 | 8.70 | 8.11 | 21.94 |
| $MapAnything_{ghost}$ | **1.54** | **1.24** | **7.02** | **7.59** | **22.60** |
| $\pi^3$ | 1.60 | 1.34 | 7.44 | 8.27 | 61.60 |
| $\pi^3_{ghost}$ | **1.43** | **1.20** | **6.70** | **7.89** | **75.26** |
| VGGT | 0.76 | 2.81 | 1.78 | 7.18 | 57.28 |
| $VGGT_{ghost}$ | **0.45** | **0.64** | **0.55** | **4.96** | **86.72** |

is synthesized through alpha blending:

$$\hat{\mathbf{I}} = \hat{\boldsymbol{\alpha}} \odot \hat{\mathbf{F}} + (1 - \hat{\boldsymbol{\alpha}}) \odot \mathbf{B} \qquad (7)$$

where $\odot$ denotes the Hadamard product. This formulation allows us to train the network by minimizing the reconstruction loss $\mathcal{L}_{rec} = ||\hat{\mathbf{I}} - \mathbf{I}||_1$, forcing the network to disentangle the alpha and foreground components solely through the physics of light transport.

**Training Strategy.** The training of TransDecomp is supervised by a compound loss function designed to ensure both physical consistency and spatial coherence. The total objective $\mathcal{L}_{total}$ is a weighted sum of a reconstruction term and a smoothness regularization term:

$$\mathcal{L}_{total} = \lambda_{rec}\mathcal{L}_{rec} + \lambda_{smooth}\mathcal{L}_{tv} \qquad (8)$$

where $\lambda_{rec}$ and $\lambda_{smooth}$ are hyper-parameters balancing the two objectives. We set $\lambda_{rec} = 5.0$ and $\lambda_{smooth} = 0.5$ to prioritize reconstruction quality while maintaining structural coherence. To validate the decomposition, the synthesized image $\hat{\mathbf{I}}$ must match the original input $\mathbf{I}$ within the object

region. We employ a masked $L_1$ loss, normalized by the object area to be invariant to object scale:

$$\mathcal{L}_{rec} = \frac{1}{\sum_{i,j} \mathbf{M}_{i,j}} \sum_{i,j} \mathbf{M}_{i,j} \cdot \left| \hat{\mathbf{I}}_{i,j} - \mathbf{I}_{i,j} \right| \qquad (9)$$

This term forces the network to accurately predict $\hat{\boldsymbol{\alpha}}$ and $\hat{\mathbf{F}}$ such that they satisfy the alpha blending equation (Eq. 7). Transparent objects often exhibit smooth surface reflections or diffuse components. To prevent high-frequency background textures from leaking into the foreground prediction $\hat{\mathbf{F}}$, we impose a Total Variation constraint. This minimizes the gradients in both horizontal and vertical directions:

$$\mathcal{L}_{tv} = \frac{1}{N} \sum_{i,j} \left( \left| \hat{\mathbf{F}}_{i+1,j} - \hat{\mathbf{F}}_{i,j} \right| + \left| \hat{\mathbf{F}}_{i,j+1} - \hat{\mathbf{F}}_{i,j} \right| \right) \quad (10)$$

This regularization encourages piecewise smoothness in the estimated foreground, effectively decoupling the sharp background edges from the object's intrinsic appearance.

### 3.4. Surface Normal Estimation

**DINO-Aligned Fusion Network for Normal Estimation.** To recover fine-grained surface normals from transparent objects, we propose DAF-Net, an architecture that bridges the gap between semantic priors and local geometric cues. As illustrated in Fig.1, DAF-Net incorporates DINOv3 as a frozen semantic branch and introduces a Multi-Modal Spatial Adapter (MMSA) to recover high-frequency spatial details.

**Multi-Modal Spatial Adapter.** Since transformer features often lack precise edge information due to patch embedding, the MMSA encoder is designed to extract pixel-aligned geometric features. It takes a concatenated tensor $X_{in} \in \mathbb{R}^{H \times W \times 6}$ as input, comprising RGB, binary mask, alpha matte, and foreground guidance. MMSA employs a hierarchical CNN structure to generate multi-scale feature maps $C_l$, where $l \in \{1, 2, 3, 4\}$, aligning spatially with the DINO features at the bottleneck.

**Fidelity-Aware Fusion Decoder.** We propose a Fidelity-Aware Fusion Block, to effectively merge the semantic context from DINOv3 with the spatial details from MMSA. At each decoding stage $l$, the upsampled features $F_{up}$ are concatenated with the corresponding skip connection $C_l$. Crucially, to adaptively suppress noise from transparent regions, we integrate a Squeeze-and-Excitation (SE) mechanism $\mathcal{A}_{se}(\cdot)$ to re-calibrate channel-wise feature responses:

$$F_l = \mathcal{A}_{se} \left( \mathcal{F}_{conv} \left( [\text{Upsample}(F_{l+1}), C_l] \right) \right) \qquad (11)$$

where $\mathcal{F}_{conv}$ denotes the convolution operation, and $[\cdot, \cdot]$ represents concatenation. The SE mechanism computes channel weights via global average pooling followed by a

*Table 5.* Ablation study on different module configurations using MoGe2 as the downstream baseline. We compare the original model, partial configurations (TDINO=TransDINO, TDecomp=TransDecomp, DAF=DAF-Net and GSTNet=GeoSemTransNet), and the full GHOST pipeline.

| Method / Configuration | Depth Metrics | | | | | | | | Normal Metrics | | | | | |
|---|---|---|---|---|---|---|---|---|---|---|---|---|---|---|
| | RMSE $\downarrow$ | REL $\downarrow$ | MAE $\downarrow$ | $\delta_{1.01} \uparrow$ | $\delta_{1.03} \uparrow$ | $\delta_{1.05} \uparrow$ | $\delta_{1.10} \uparrow$ | $\delta_{1.25} \uparrow$ | Mean $\downarrow$ | Median $\downarrow$ | RMSE $\downarrow$ | $11.25° \uparrow$ | $22.5° \uparrow$ | $30° \uparrow$ |
| MoGe2 (Original) | 0.018 | 0.025 | 0.014 | 0.325 | 0.711 | 0.867 | 0.972 | 0.999 | 30.03 | 26.06 | 35.64 | 17.50 | 45.24 | 60.00 |
| *w/* TDINO + GSTNet | 0.089 | 0.049 | 0.075 | 0.093 | 0.333 | 0.576 | 0.745 | 0.800 | 63.24 | 53.23 | 57.92 | 5.32 | 26.44 | 39.88 |
| *w/* TDINO + DAF + GSTNet | 0.074 | 0.040 | 0.071 | 0.143 | 0.318 | 0.626 | 0.835 | 0.859 | 54.58 | 48.34 | 49.87 | 7.75 | 33.35 | 42.35 |
| *w/* TDINO + TDecomp + GSTNet | 0.045 | 0.356 | 0.039 | 0.252 | 0.473 | 0.800 | 0.874 | 0.928 | 50.35 | 43.62 | 45.97 | 16.24 | 41.43 | 64.33 |
| **MoGe2 + GHOST (Full Pipeline)** | **0.014** | **0.018** | **0.010** | **0.433** | **0.820** | **0.933** | **0.990** | **1.000** | **19.74** | **16.35** | **24.37** | **32.87** | **68.35** | **81.44** |

*Table 6.* Ablation study on 3D reconstruction metrics using $\pi^3$ as the baseline. We evaluate the contribution of different geometric priors in the GHOST pipeline. $\alpha$, $\beta$, and $\gamma$ denote the ablation sequence for the depth estimation task.

| Method | Acc $\downarrow$ | Comp $\downarrow$ | CD $\downarrow$ | ND $\downarrow$ | F-Score $\uparrow$ |
|---|---|---|---|---|---|
| $\pi^3$ (Original) | 1.60 | 1.34 | 7.44 | 8.27 | 61.61 |
| *w/$\alpha$* | 3.28 | 2.20 | 10.23 | 9.61 | 26.35 |
| *w/$\beta$* | 3.07 | 2.06 | 9.50 | 9.57 | 31.36 |
| *w/$\gamma$* | 2.44 | 1.87 | 9.11 | 8.81 | 39.83 |
| $\pi^3_{GHOST}$ | **1.43** | **1.20** | **6.70** | **7.89** | **75.26** |

bottleneck MLP, explicitly modeling the inter-dependencies between semantic and geometric channels.

To jointly optimize the surface normal accuracy and the estimated uncertainty, we formulate the training objective as a weighted sum of a probabilistic reconstruction term and a geometric regularization term:

$$\mathcal{L} = \lambda_{\text{vmf}}\mathcal{L}_{\text{vmf}} + \lambda_{\text{smooth}}\mathcal{L}_{\text{smooth}} \quad (12)$$

**Probabilistic Reconstruction Loss.** We model predicted normals via the Von Mises-Fisher (vMF) distribution. To resolve orientation ambiguity (inward vs. outward surfaces), we employ a sign-agnostic Negative Log-Likelihood (NLL) loss. For each pixel $p$ within mask $\Omega$, the loss is:

$$\mathcal{L}_{\text{vmf}} = \frac{1}{|\Omega|}\sum_{p \in \Omega}\left(\hat{\kappa}_p\left(1 - \left|\hat{\mathbf{n}}_p \cdot \mathbf{n}_p^{\text{gt}}\right|\right) - \log\hat{\kappa}_p\right) \quad (13)$$

where $\hat{\mathbf{n}}_p$ and $\mathbf{n}_p^{\text{gt}}$ are the predicted and ground-truth unit normals, and $\hat{\kappa}_p$ is the predicted concentration parameter. The term $-\log\hat{\kappa}_p$ acts as a regularizer to prevent the uncertainty from collapsing, while the cosine distance is weighted by the confidence $\hat{\kappa}_p$.

**Edge-Aware Smoothness Prior.** Transparent objects feature smooth surfaces with sharp silhouettes. We enforce spatial consistency via an edge-aware smoothness loss guided by $\alpha_{trans}$, encouraging normals to vary smoothly in flat regions while preserving discontinuities at geometric boundaries:

$$\mathcal{L}_{\text{smooth}} = \sum_{d \in \{x,y\}} \mathbb{E}\left[\exp\left(-\gamma\left|\nabla_d\mathbf{A}\right|\right) \cdot \left|\nabla_d\hat{\mathbf{n}}\right|\right] \quad (14)$$

Here, $\nabla_d$ denotes the spatial gradient in direction $d$, $\mathbf{A}$ represents the input alpha matte, and $\gamma$ controls the sensitivity to boundary edges.

### 3.5. Structure-Preserving Opaque Texture Synthesis

We propose GeoSemTransNet, a generative framework leveraging semantic consistency and geometric priors. A texture encoder $E_{tex}$ extracts features $F_{tex}$. To supplement transparent objects' weak textures, a semantic adapter projects frozen DINOv3 features $F_{dino}$ into the encoder space via $1 \times 1$ convolution and GroupNorm. The fused latent code is $Z = E_{tex}(I_{src}) + \psi(F_{dino})$, with $\psi$ as the projection head. Decoding is guided by priors $\mathcal{P} = \{\mathbf{N}, \mathbf{M}, I_{fg}\}$ from TransDecomp. We employ Spatially-Adaptive Normalization (SPADE) blocks rather than concatenation; for feature map $h$ at layer $l$, the activation is modulated as:

$$h'_{n,c,y,x} = \gamma_{c,y,x}(\mathcal{P}) \cdot \frac{h_{n,c,y,x} - \mu_c}{\sigma_c} + \beta_{c,y,x}(\mathcal{P}) \quad (15)$$

where $\gamma$ and $\beta$ are learned modulation parameters derived specifically from the geometric structure. This ensures that the restored texture strictly adheres to the object's 3D geometry.

**Physics-Aware Loss Functions.** We introduce a physics-based loss landscape to constrain the generation. (1) Shape-from-Shading Constraint: We enforce the luminance of the generated image $I_{gen}$ to be consistent with the diffuse shading implies by the predicted surface normals $\mathbf{N}$. Assuming a dominant light source vector $\mathbf{l}$, the loss is defined as:

$$\mathcal{L}_{sfs} = \|\mathcal{T}_{gray}(I_{gen}) \odot \mathbf{M} - \text{ReLU}(\langle\mathbf{N}, \mathbf{l}\rangle) \odot \mathbf{M}\|_1 \quad (16)$$

(2) Gradient Consistency: To align textural edges with geometric discontinuities, we minimize the divergence between image gradients and normal gradients:

$$\mathcal{L}_{grad} = \|\nabla I_{gen} - \nabla\mathbf{N}\|_1 \quad (17)$$

Total Objective. The full loss function combines the physics-based constraints with pixel-wise reconstruction terms (boundary and foreground consistency):

$$\mathcal{L}_{total} = \lambda_{sfs} \cdot \mathcal{L}_{sfs} + \lambda_{grad} \cdot \mathcal{L}_{grad} + \lambda_{fg} \cdot \mathcal{L}_{fg} + \lambda_{bound} \cdot \mathcal{L}_{bound} \quad (18)$$

Empirically, we prioritize geometric fidelity by setting higher weights for the physics-consistent terms: $\lambda_{sfs} = 10.0$ and $\lambda_{grad} = 5.0$, while the auxiliary reconstruction weights are set to $\lambda_{fg} = 2.0$ and $\lambda_{bound} = 1.0$. This configuration forces the network to respect the underlying 3D shape derived from TransDecomp rather than overfitting to ambiguous transparent textures.

## 4. Experiments

We implement our model on a single A100 GPU for 70 epochs with a batch size of 8. The learning rate is governed by a cosine decay strategy ranging from $10^{-3}$ to $10^{-4}$. Input images are resized to $256 \times 256$ for both training and evaluation. Models inference in this paper is conducted on an NVIDIA GeForce RTX 4090, where GHOST processes each view in 0.38 seconds.

### 4.1. Datasets

We evaluate GHOST on ClearGrasp, ClearPose, and Trans10K-v2. ClearGrasp provides synthetic and real data with RGB, mask, depth, and surface normal annotations. ClearPose is a multi-view real-world dataset with similar modalities. Conversely, Trans10K-v2 focuses on semantic segmentation and lacks 3D geometric labels. Therefore, while all three datasets train the segmentation component, subsequent modules and downstream performance evaluations utilize only ClearGrasp and ClearPose.

### 4.2. Segmentation

We train TransDINO using official splits of ClearGrasp, ClearPose, and Trans10K-v2, and evaluate it on their respective test sets. For the multi-view ClearPose dataset, we sample every 500th frame to create a subset of 54,454 training and 2,605 testing samples. We further test generalization on the TROS transparent subset using the mean Intersection over Union (mIoU) metric. Table 1 compares TransDINO against specialized models and the visual comparison results are shown in Figure 5; notably, these baselines are trained on at most two datasets, and several require depth priors. Results show TransDINO's superior cross-domain adaptability and generalization compared to RGB and RGB-D competitors.

### 4.3. Service for Depth Estimation

**Metrics.** We evaluate depth estimation using RMSE, REL, MAE, and threshold accuracy $\delta < \{1.01, 1.03, 1.05, 1.10, 1.25\}$. For surface normals—derived from point clouds back-projected via depth and camera intrinsics—we use Mean, Median, and RMSE angular errors, plus pixel percentages within $11.25°$, $22.5°$, and $30°$. All metrics are computed exclusively within the GT transparent object masks.

**Details.** Original ClearGrasp RGB images and GHOST-processed counterparts are fed into SOTA depth estimation models. Using predicted depth and camera intrinsics, we reconstruct 3D point clouds and calculate angular errors against GT surface normals, treated as unoriented. Table 2 shows downstream performance before and after GHOST and Table 7 reports the zero-shot generalization results of GHOST on the TROS dataset; Table 3 benchmarks GHOST against specialized transparent object depth estimators. Results confirm that GHOST's opaquification enables reliable estimation via generic models. Qualitative comparisons are in Fig. 4. Figure 6 illustrates the qualitative results of depth estimation, comparing GHOST equipped with DA3 against other existing methods and workflows.

### 4.4. Service for Feed-Forward Reconstruction

**Metrics.** We evaluate the quality of feed-forward 3D reconstruction using Accuracy (Acc), Completeness (Comp), Chamfer Distance (CD), Normal Distance (ND), and F-Score. All metrics are computed exclusively within the masked regions defined by the GT transparent object masks.

**Details.** To evaluate GHOST, we select SOTA feed-forward reconstruction foundation models as baselines, comparing performance before and after pre-processing. On the multi-view ClearPose test set, we sample every 500th frame to extract sparse views. Table 4 and Fig. 4 provide quantitative and qualitative results, respectively. Findings show that GHOST-processed images enable effective surface reconstruction for originally transparent objects. Figure 7 presents the qualitative results of monocular depth estimation and feed-forward reconstruction using GHOST-processed data in one in-the-wild scene.

### 4.5. Ablation Study

**Full Pipeline Definition:** GHOST = TransDINO + TransDecomp + DAF-Net + GeoSemTransNet.

**TransDINO.** Explicitly delineating regions via masks is a prerequisite. As reliable segmentation is foundational, we omit the TransDINO-specific ablation.

**TransDINO + GeoSemTransNet.** Lacking foreground references and surface normals, GeoSemTransNet uses only masks as geometric input. This results in uniform noise-like outputs and training non-convergence, proving this configuration ineffective.

**TransDINO + DAF-Net + GeoSemTransNet.** Here, DAF-Net predicts normals without foreground/alpha cues, while GeoSemTransNet remains conditioned solely on masks. This setup yields methodologically invalid and poor results.

**TransDINO + TransDecomp + GeoSemTransNet.** Rely-

ing exclusively on masks and foreground maps, this configuration produces suboptimal performance and detrimental artifacts that hinder downstream tasks.

**Quantitative Analysis.** We evaluate these settings using the established depth and reconstruction metrics, selecting MoGe v2 and Pi3 as representative downstream baselines. As shown in Table 5 and Table 6, other module combinations generate opaque surfaces that downstream models interpret as noise. Consequently, their prediction quality is often inferior to that of the original raw images, underscoring the necessity of the full GHOST pipeline.

# 5. Conclusion

We present GHOST, a preprocessing framework that transforms transparent regions into opaque surfaces to overcome 3D perception failures and high training costs. By sequentially predicting segmentation masks, foreground grayscale maps, alpha mattes, and surface normals, GHOST synthesizes structurally consistent opaque textures. Experiments show that GHOST enables off-the-shelf depth estimators and reconstruction networks to achieve accurate inference on transparent objects and provides a new paradigm for resolving transparent object bottlenecks in general 3D vision tasks.

# Acknowledgements

We sincerely thank the anonymous reviewers for their valuable comments and constructive suggestions, which have significantly improved the quality of this paper. We also thank the organizers and program committee of ICML for arranging the review process. This work was supported by the National Natural Science Foundation of China under Grant No. 52130403.

# Impact Statement

This paper presents work whose goal is to advance the field of Machine Learning. There are many potential societal consequences of our work, none which we feel must be specifically highlighted here.

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

# A. Appendix

*Table 7.* Comparison of the performance between the monocular depth estimation model and the preprocessed monocular depth estimation model on the unseen TROS dataset. $\delta$ denotes threshold accuracy, while angular errors are measured in degrees.

| Method | Depth Metrics | | | | | | | | Normal Metrics | | | | | |
| --- | --- | --- | --- | --- | --- | --- | --- | --- | --- | --- | --- | --- | --- | --- |
| | RMSE ↓ | REL ↓ | MAE ↓ | $\delta_{1.01}$ ↑ | $\delta_{1.03}$ ↑ | $\delta_{1.05}$ ↑ | $\delta_{1.10}$ ↑ | $\delta_{1.25}$ ↑ | Mean ↓ | Median ↓ | RMSE ↓ | 11.25° ↑ | 22.5° ↑ | 30° ↑ |
| DA3 | 0.174 | 0.037 | 0.121 | 0.271 | 0.614 | 0.779 | 0.921 | 0.991 | 36.84 | 32.01 | 43.12 | 14.33 | 39.11 | 53.04 |
| DA3 + GHOST | **0.135** | **0.028** | **0.094** | **0.362** | **0.731** | **0.861** | **0.956** | **0.995** | **24.26** | **19.88** | **29.77** | **27.89** | **61.02** | **74.76** |
| MoGe2 | 0.137 | 0.030 | 0.101 | 0.300 | 0.677 | 0.835 | 0.950 | 0.996 | 33.97 | 29.53 | 40.21 | 15.97 | 42.13 | 56.77 |
| MoGe2 + GHOST | **0.106** | **0.016** | **0.067** | **0.407** | **0.789** | **0.904** | **0.977** | **0.998** | **22.34** | **18.66** | **27.49** | **30.41** | **64.11** | **77.88** |
| DepthPro | 0.143 | 0.032 | 0.108 | 0.288 | 0.652 | 0.811 | 0.941 | 0.996 | 32.71 | 28.22 | 38.76 | 16.72 | 44.22 | 59.01 |
| DepthPro + GHOST | **0.134** | **0.029** | **0.093** | **0.329** | **0.698** | **0.844** | **0.957** | **0.997** | **28.23** | **24.31** | **33.65** | **19.67** | **50.66** | **66.43** |
| Metric3Dv2 | 0.211 | 0.044 | 0.152 | 0.193 | 0.507 | 0.701 | 0.900 | 0.989 | 48.56 | 46.12 | 54.98 | 6.61 | 21.45 | 32.77 |
| Metric3Dv2 + GHOST | **0.158** | **0.033** | **0.117** | **0.249** | **0.619** | **0.802** | **0.951** | **0.995** | **42.99** | **39.77** | **49.44** | **9.68** | **28.11** | **40.66** |
| VGGT | 0.157 | 0.033 | 0.109 | 0.307 | 0.669 | 0.817 | 0.932 | 0.992 | 35.22 | 30.61 | 41.97 | 17.21 | 42.44 | 55.51 |
| VGGT + GHOST | **0.149** | **0.026** | **0.088** | **0.400** | **0.754** | **0.876** | **0.961** | **0.994** | **22.87** | **18.93** | **28.33** | **31.09** | **63.22** | **76.21** |

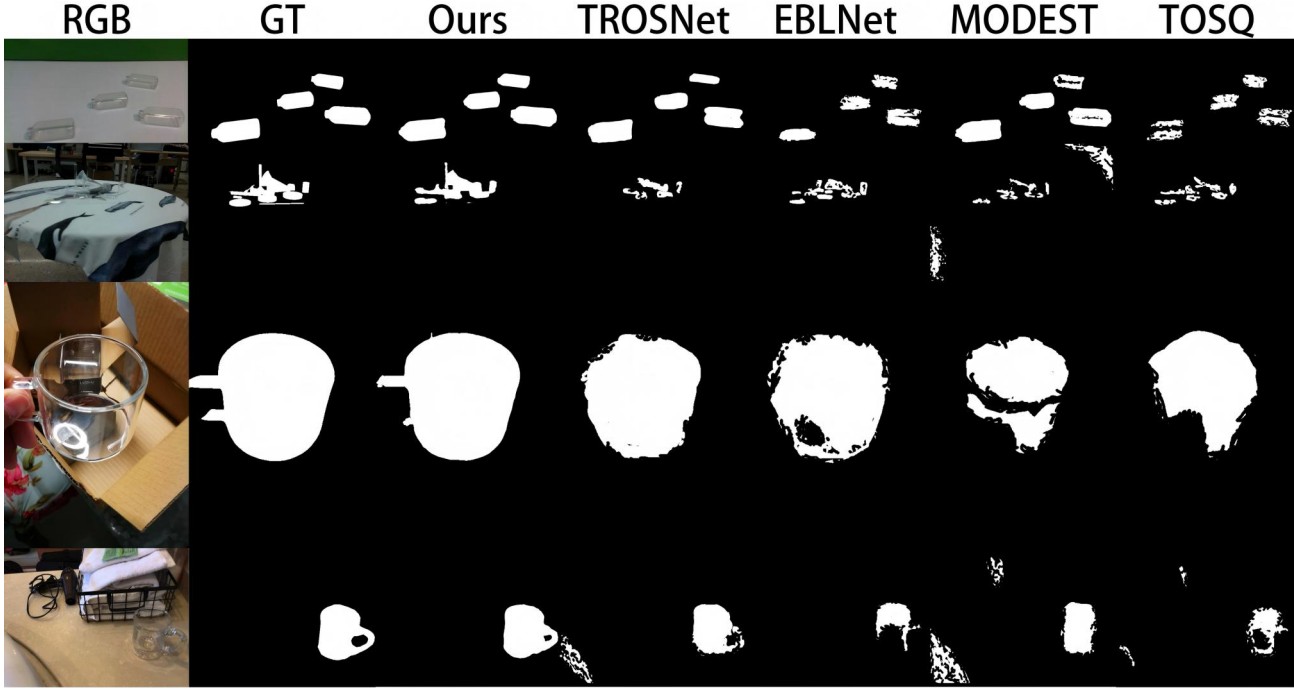

*Figure 5.* Visualization of segmentation results of TransDINO (Ours) and other transparent object segmentation models with high test accuracy on views from different datasets.

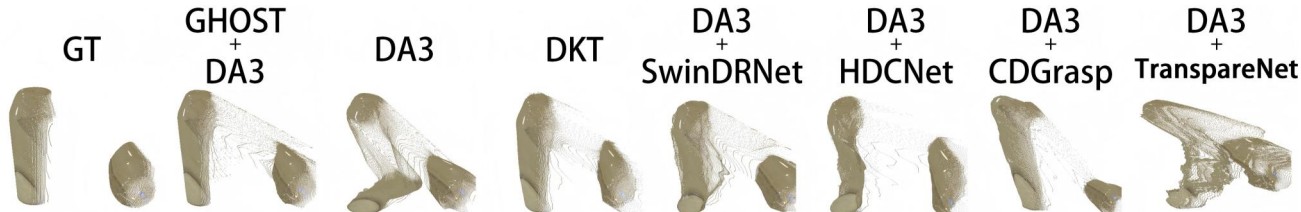

*Figure 6.* Visualization of 3D point cloud mapping results (via camera intrinsic parameters) for four depth estimation outputs: depth predicted by DepthAnythingV3 with GHOST preprocessing, depth directly predicted by DA3, depth predicted by DKT (a dedicated model for transparent objects), and depth processed by various depth completion models using DA3's predicted depth as input.

*Figure 7.* Visualization of 3D point clouds derived from monocular depth estimation (via camera intrinsic parameters) using different models on a single real-world view, and single-view reconstruction directly performed by feedforward reconstruction models. Each method is compared with the counterpart that adopts GHOST as preprocessing.

