# OpenReview forum: "GHOST: Geometry-Guided Hallucination of Opaque Surface Textures"
_ICML.cc/2026/Conference — ICML 2026 regular_

### Official Review · Reviewer_7pPc · 2026-03-11

**Soundness:** 3
**Presentation:** 3
**Significance:** 3
**Originality:** 3
**Overall Recommendation:** 4
**Confidence:** 3

**Summary:**

This paper introduces a new framework to improve the geometric estimation of transparent objects. The framework, GHOST, comprises four distinct modules. The pipeline works as follows. TransDINO estimates a segmentation mask, which TransDecomp uses to predict alpha and foreground. DAF-Net then estimates the normal and confidence with the output of TransDecomp. Finally, GeoSemTransNet synthesizes opaque textures. In the experiments, the evaluation includes depth estimation, normal estimation, and 3D reconstruction. The authors use renowned models as baseline and show that the framework actually improve the performance.

**Compliance With Llm Reviewing Policy:**

Affirmed.

**Final Justification:**

The rebuttal addressed my concerns.

**Key Questions For Authors:**

The overall paper is good to me. I would raise my score if the questions asked above is answered. Thank the author for their hard work.

**Limitations:**

I do not see a specific section discussed about limitation. I believe the research has no negative impact.

**Strengths And Weaknesses:**

Strengths
* The paper introduces a novel approach for transparent objects geometry understanding.
* The framework comprises four novel modules, each of them are explained in detail. TransDINO even has its own assessment.
* The evaluation includes depth estimation, normal estimation, and 3D reconstruction. The baseline includes many SOTA methods, and the results validates GHOST's effectiveness.

Weakness
* In the experiments, the size of each image is 256x256. This is weird to me since every SOTA method in your baseline uses higher resolution. It also makes me question the effectiveness of TransDINO and TransDecomp. Why do you choose low-resolution setting? Is this general in transparent object related tasks? Can TransDINO and TransDecomp be replaced with SOTA matting model?

* What is the size of the dataset, especially test set? How many of them are real world images? Can your model generalize to other dataset? This questions should be explained.

---

> ### Author Rebuttal · Authors · 2026-03-26
>
> We sincerely thank you for the constructive feedback and the high recognition of our work. Below, we address your specific questions in detail.
> ### **Q1. Choice of Image Resolution**
> We apologize for any confusion regarding our resolution settings.
> - **Pre-processing Resolution:** The 256×256 resolution mentioned in Section 4 refers specifically to the GHOST pre-processing framework itself. We chose this setting to demonstrate the framework's efficiency, achieving a real-time processing speed of 0.38s per view on an RTX 4090.
> - **Downstream Integration:** For the qualitative and quantitative analysis of monocular depth estimation and feed-forward reconstruction, the new RGB views supplied to downstream models are actually at 512×512 resolution. We apologize for omitting this crucial detail in the manuscript.
> - **Compatibility with SOTA Models:** SOTA models like DepthAnythingV3 and VGGT utilize DINO-based backbones with a 14×14 pixel patch size. To avoid zero-padding or interpolation residuals, input dimensions must be multiples of 14 (target sizes like 504 or 518). GHOST is designed as a universal framework; even when processing at 256×256, the resulting opaque textures can be upsampled to 512×512 or other model-specific target sizes while maintaining reliable geometric consistency.
> - **Practical Considerations:** In robotic manipulation or embedded systems, 256×256 is a standard target resolution that balances accuracy with real-time requirements. By using this as our baseline description, we ensure GHOST remains compatible with a wide range of hardware and downstream methods.
>
> ### **Q2. Distinguishing GHOST from Matting Models**
> We appreciate your question on whether TransDINO and TransDecomp could be replaced by SOTA matting models. While we adopt an alpha blending method, our task objective differs fundamentally from image matting:
> - **Physical Decomposition vs. Visual Extraction:** Standard matting models prioritize visual seamlessness and aesthetic synthesis. In contrast, TransDecomp is a physics-constrained module. Its goal is to disentangle surface physical properties into transparency $\alpha$ and foreground reflection $F_g$ to provide pure signals for 3D recovery.
> - **3D Structural Consistency:** In the GHOST pipeline, these outputs are not final results but serve as geometric and texture cues for DAF-Net and GeoSemTransNet. Existing matting models lack the structured physical modeling required to provide the geometrically consistent intermediate representations needed for 3D tasks.
> - **Handling Refractive Ambiguity:** TransDINO utilizes DINOv3 semantic priors** to robustly locate boundaries where internal textures are absent. TransDecomp employs Gated Convolutions to distinguish between surface reflections and transmitted background patterns —a level of physical decoupling that general-purpose matting models cannot achieve.
> In summary, TransDINO and TransDecomp are physical property extractors rather than image cutters, ensuring the synthesized "opaque textures" are accurately recognizable by 3D foundation models.
>
> ### **Q3. Dataset Scale and Model Generalization**
> Thank you for the reminder to provide more detail on our dataset usage. We utilized four datasets: ClearGrasp, ClearPose, Trans10K-v2 and TROS, adhering to official splits where available.
>
> | Dataset     | Training | Modules Involved  | Training Size/Real | Validation/Real | Test/Real |
> | ----------- | -------- | ----------------- | ------------------ | --------------- | --------- |
> | ClearGrasp  | √        | Complete Pipeline | 45453/0            | 719/173         | 507/113   |
> | ClearPose   | √        | Complete Pipeline | 4000/4000          | 400/400         | 64/64     |
> | Trans10K-v2 | √        | TransDINO         | 5000/5000          | 800/800         | 200/200   |
> | TROS        | ×        | Complete Pipeline | 0                  | 0               | 3385/3385 |
>
> - ClearGrasp: Contains real data only in validation/test sets; used for evaluating monocular depth estimation.
> - ClearPose: A video-based dataset where we sampled every 5th frame for training to match the single-view format of ClearGrasp. For 3D reconstruction testing, we utilized 12 groups of scenes, sampling every 500th frame to create sparse view sets for the reconstruction models.
> - Trans10K-v2: Used exclusively for training and testing the segmentation performance of TransDINO.
> - TROS: To demonstrate zero-shot generalization, GHOST was tested on the TROS dataset without any prior training on this data. While the manuscript originally focused on TransDINO's performance on TROS, we have now added the GHOST+Monocular Depth results in Table 1 of `rebuttal.pdf` in our anonymous repository [https://anonymous.4open.science/r/GHOST-3B50/](https://anonymous.4open.science/r/GHOST-3B50/).
>
> We have also provided new visual evidence in the repository: Figure 1:TransDINO segmentation, Figure 2:GHOST vs. depth completion and Figure 3:in-the-wild visualization.

---

> > ### Author Rebuttal · Reviewer_7pPc · 2026-04-03
> >
> > I appreciate the author's efforts in the rebuttal, and my concerns have been addressed. Thank you.

---

> > > ### Author Response · Authors · 2026-04-04
> > >
> > > Thank you for your response and for confirming that our clarifications have fully addressed your concerns. We are pleased that our efforts during the rebuttal period met your expectations.
> > >
> > > We are sincerely grateful for the professional guidance you have provided throughout this process. As the discussion period draws to a close, we hope the current standing of the work reflects the quality you were looking for to support its progress in the final assessment. Thank you again for your time and for the care you have dedicated to our work.

---

### Official Review · Reviewer_wJku · 2026-03-11

**Soundness:** 2
**Presentation:** 3
**Significance:** 3
**Originality:** 2
**Overall Recommendation:** 4
**Confidence:** 3

**Summary:**

his work introduces GHOST, a geometry-aware preprocessing framework designed to handle transparent objects in visual reconstruction pipelines. Instead of modifying existing reconstruction networks, the method converts transparent regions into opaque RGB representations, enabling standard depth estimation and feed-forward reconstruction models to operate on them directly without additional training. The framework includes four main components: TransDINO, which detects transparent-object regions; TransDecomp, which separates transparency effects from foreground appearance; DAF-Net, responsible for estimating surface normals; and GeoSemTransNet, which generates opaque textures that remain consistent with the underlying geometry. Experimental results demonstrate that the system produces plausible textures for transparent areas, leading to improved performance in downstream tasks such as segmentation, depth prediction, and normal estimation.

**Compliance With Llm Reviewing Policy:**

Affirmed.

**Final Justification:**

I appreciate the author's efforts in the rebuttal, and I think the additional results address my concerns. I would therefore like to raise my score and encourage the authors to include these further analyses and results in the revision to strengthen the paper.

**Key Questions For Authors:**

1. Experimental setup and fairness. Tables 1 and 3 are the most informative comparisons in the paper, while the other experiments mainly focus on applications with off-the-shelf models. Could the authors also evaluate depth estimation on other datasets? In addition, for baselines that require RGB-D input (e.g., for HDCNet), it is unclear what depth input is used. More details about the experimental setup would help clarify whether the comparisons are fair. It may also be beneficial to include direct qualitative comparisons for both the segmentation and depth estimation tasks against baseline methods, rather than mainly showing improvements when combined with other depth models (e.g., GHOST vs. GHOST + VGGT).
2. Dependency of Priors DINOv3. As mentioned in the paper, both TransDINO and DAF-Net rely on DINOv3 priors. The authors also note that DINOv3 already captures useful cues for transparent objects, which may contribute to depth and normal estimation. It is unclear how much of the performance gain comes from the proposed pipeline versus the use of DINOv3 features. Could the authors provide an ablation study replacing DINOv3 with earlier features (e.g., DINOv2) while keeping the rest of the architecture unchanged?
3. How well does the proposed method generalize to in-the-wild data? It would also be helpful if the authors could discuss potential failure cases of the method to better understand its limitations.

**Limitations:**

It would be helpful if the authors could place more emphasis on evaluating the quality of the generated opaque surface textures (e.g., compared to diffusion-based baselines) and include more direct comparisons with existing depth correction or completion baselines.

**Strengths And Weaknesses:**

**Strength**
1. Significance. The paper tackles a practically important problem: transparent objects often break standard 3D perception pipelines due to reflection and refraction effects. The proposed method aims to recover plausible textures for transparent regions so that existing depth estimation and reconstruction models can operate more reliably.
2. Presentation. The paper is  well written and clearly structured. The four components are organized in a clean pipeline with an accompanying figure, which makes the overall method easy to follow.
3. An appealing aspect of the proposed approach is that it can be applied as a pre-processing module for existing off-the-shelf depth estimation and reconstruction models without requiring retraining. The experiments demonstrate that integrating GHOST with these models can consistently improve their performance on scenes containing transparent objects.

**Weakness**
1. Soundness. While the paper evaluates the proposed method on several downstream tasks and shows improvements, the comparison with methods designed for depth completion or image inpainting for transparent objects is limited. It is not particularly surprising that introducing a pre-processing module improves the performance of off-the-shelf depth estimation models. The current experiments focus heavily on the application results when combined with pretrained 3D foundation models. However, a more thorough evaluation with stronger baselines and additional datasets for depth correction would be helpful, as the authors also didn’t directly evaluate the quality of the generated images themselves.
2. Originally. The pipeline consists of several relatively complex modules, which makes the core contribution somewhat difficult to evaluate. As mentioned by the authors, DINOv3 features already capture useful cues for transparent objects such as boundaries and structural features. It remains unclear how much of the final performance gain comes specifically from the proposed components versus the use of strong pretrained features. Moreover, GHOST is a fairly heavy four-stage pipeline with multiple separately designed modules and priors. While the full system appears effective, it is difficult to determine which components are truly essential and which mainly contribute engineering complexity.
3. Generalization. The paper does not sufficiently evaluate the generalization ability of the method for opaque surface textures in different scenarios. It would also be helpful to analyze potential failure cases to better understand the limitations of the approach.

---

> ### Author Rebuttal · Authors · 2026-03-25
>
> We appreciate your insightful comments regarding our framework's evaluation and architecture. Below are our direct responses to the points you raised:
>
> ---
>
> ### **Q1. Experimental Setup and Fairness**
>
> We thank **you** for pointing out the need for depth estimation evaluation on additional datasets.
>
> - **New Evaluations:** To address **your** concern, we have supplemented the performance results of our GHOST-integrated monocular depth estimation models on the **TROS dataset**, which was not used during our training phase. These results are provided in **Table 1** of `rebuttal.pdf` in our anonymous repository https://anonymous.4open.science/r/GHOST-3B50/.
>
> - **Baseline Input Details:** We apologize that space constraints in the original manuscript prevented a detailed description of the baseline experimental settings. For the depth completion baselines—**HDCNet** , **SwinDRNet** , **TranspareNet** , and **ClueDepthGrasp** —the input is **RGB-D**. The **D-channel** consists of depth maps obtained directly from sensor hardware or predicted by standard depth estimation models. As we noted in the paper, sensors typically fail to capture accurate depth for transparent objects due to refraction and reflection, leading to inaccurate or missing values.
>
> - **Qualitative Analysis:** We appreciate **your** reminder to include qualitative comparisons. Our initial focus was on quantifying 3D task improvements, but we agree that visual evidence is essential. We have now added side-by-side visual comparisons for **2D segmentation** (**Figure 1**, `rebuttal.pdf`) and **depth estimation** (**Figure 2**, `rebuttal.pdf`) between our pipeline and other depth completion methods in the anonymous repository.
>
>
> ---
>
> ### **Q2. Dependency of Priors: DINOv3**
>
> We thank **you** for the suggestion to evaluate earlier features like **DINOv2**. This is a valuable way to verify GHOST’s decoupling from specific foundation models.
>
> 1. **Complexity and Time Constraints:** To be candid, replacing DINOv3 with DINOv2 requires a full retraining and re-evaluation of three core components: **TransDINO** (segmentation) , **DAF-Net** (normals) , and **GeoSemTransNet** (synthesis). Since GHOST is a tightly coupled multi-stage pipeline , completing full-scale retraining across all datasets while ensuring experimental quality is extremely challenging within the one-week rebuttal window.
>
> 2. **Architecture vs. Feature Strength:** We want to emphasize that while DINOv3 provides superior structural and boundary priors, our core contribution is the **GHOST framework’s topology and fusion mechanism**. As shown in our ablation studies (**Table 5 and Table 6**) , strong foundation features alone are insufficient; without our **MMSA adapter** for spatial detail recovery or the **SPADE-based GeoSemTransNet** for geometric conditioning , downstream models still interpret processed images as noise. This proves that the performance gain stems from how GHOST **organizes and structurally fuses** these priors.
>
> 3. **Future Commitment:** We fully agree that verifying scalability is vital. We have initiated follow-up optimization using **DINOv2 as the backbone** and commit to including these data in the final version of the manuscript.
>
>
> ---
>
> ### **Q3. Generalization on In-the-Wild Data**
>
> **Your** suggestions regarding in-the-wild generalization and failure case analysis have been very helpful. We have added visualization results before and after pre-processing in **Figure 3** of `rebuttal.pdf` in the anonymous repository.
>
> - **Limitations of the Assumption:** As we discussed, our method has limitations on surfaces that violate the **"weak refraction" assumption**. In these cases, the background pixels are not clearly visible through the object, making it appear visually "non-transparent."
>
> - **Real-world Context:** Many objects we consider transparent (e.g., solid crystal ball ornaments) do not truly allow background objects to be clearly seen through them. While GHOST may not synthesize ideal new pixels for such cases, these objects effectively behave as opaque surfaces. In such scenarios, standard depth estimation or 3D reconstruction models can already produce reliable results without additional pre-processing.
>
> We hope our detailed clarifications and the additional results in `rebuttal.pdf` effectively address your concerns regarding experimental fairness and architectural novelty. We would be grateful if you could **reconsider your evaluation** in light of these updates.

---

> > ### Author Rebuttal · Reviewer_wJku · 2026-04-03
> >
> > I appreciate the author's efforts in the rebuttal, and I think the additional results address my concerns. I would therefore like to raise my score and encourage the authors to include these further analyses and results in the revision to strengthen the paper.

---

> > > ### Author Response · Authors · 2026-04-04
> > >
> > > Thank you very much for your feedback and for confirming that your concerns have been fully resolved. We are particularly grateful for your encouragement and for your willingness to raise the score to reflect the current status of the work.
> > >
> > > We will ensure that all the additional analyses and results discussed during this rebuttal are included in the final revision to further strengthen the paper, as you suggested. Thank you again for your time and for the support you have shown for our work.

---

### Official Review · Reviewer_8oR8 · 2026-03-12

**Soundness:** 3
**Presentation:** 3
**Significance:** 3
**Originality:** 3
**Overall Recommendation:** 4
**Confidence:** 4

**Summary:**

GHOST is a preprocessing framework that transforms transparent objects into opaque surfaces with realistic textures through a four-module pipeline: TransDINO for precise segmentation using DINOv3 semantic features, TransDecomp for decomposing transparency into alpha mattes and foreground maps via physics-constrained learning, DAF-Net for estimating surface normals with geometric regularization, and GeoSemTransNet for synthesizing structure-preserving opaque textures guided by geometric priors and physics-aware losses. By converting challenging transparent regions into opaque equivalents, GHOST enables off-the-shelf depth estimators and 3D reconstruction networks to achieve accurate inference without specialized training, demonstrated through significant improvements across multiple datasets and downstream tasks while processing each view in just 0.38 seconds.

**Compliance With Llm Reviewing Policy:**

Affirmed.

**Final Justification:**

Thanks for the rebuttal. The additional explanations fully address my concerns. I will keep my score.

**Key Questions For Authors:**

(1) Could you provide more details on the "weak refraction" assumption and its limitations? Under what conditions (e.g., object thickness, curvature, material properties) does the alpha blending approximation begin to break down, and how would failure cases manifest in the downstream predictions?

(2) The ablation studies show that partial configurations (e.g., w/o TransDecomp) produce results worse than the original raw images, with downstream models interpreting the outputs as noise. Could you elaborate on what specifically causes this degradation? Is it primarily the introduction of unrealistic textures, incorrect geometry, or misaligned boundaries that confuse the downstream models?

**Limitations:**

The authors should explicitly discuss the boundaries of their approach, the "weak refraction" assumption, failure cases (e.g., thick glass, complex geometries, extreme lighting conditions), sensitivity to segmentation accuracy, and dependency on specific foundation models. A dedicated "Limitations" section would strengthen the paper.

**Strengths And Weaknesses:**

The paper demonstrates strong technical soundness through its well-justified modular architecture, rigorous mathematical formulations, and comprehensive experimental validation across multiple datasets with appropriate metrics. Its ablation studies are particularly well-designed, systematically proving that all four modules are necessary for optimal performance. The work addresses a problem of high practical significance, enabling standard 3D perception models to handle transparent objects, with a clever preprocessing approach that achieves substantial quantitative improvements (e.g., mIoU increasing from 71.40% to 92.44% on ClearGrasp) while processing each view in just 0.38 seconds, making it viable for real-world applications. The paper demonstrates strong originality through its creative synthesis of existing techniques into a unified pipeline, with particularly innovative elements including the cross-attention mechanism for fusing RGB features with DINOv3 semantic embeddings, physics-constrained decomposition trained through alpha blending, and structure-preserving texture synthesis guided by geometric priors and physics-aware losses.

---

> ### Author Rebuttal · Authors · 2026-03-24
>
> We appreciate your recognition of our work's technical soundness and the value of our modular architecture. We are happy to provide detailed clarifications regarding the physical assumptions and the experimental observations you noted.
>
> ---
>
> ### **Q1. Clarification on the "Weak Refraction" Assumption**
>
> We apologize for the brevity of this description in the main text due to space constraints.
> Physical Foundation:
> According to **Snell's Law**:
> $$n_1 \sin(\theta_1) = n_2 \sin(\theta_2)$$
> When the refractive index difference between medium 1 and medium 2 is minimal ($n_1 \approx n_2$), the incident angle $\theta_1$ and the refractive angle $\theta_2$ are nearly identical. Visually, light travels through the object in a nearly straight line with minimal bending or reflection. In such cases of **weak refraction**, the object appears highly transparent, and we can mathematically simplify the interaction as **Alpha Blending**. This allows us to ignore complex ray-tracing or geometric distortions while maintaining computational efficiency.
>
> In cases of **strong refraction** (where $n_2 \gg n_1$), the object surface appears less transparent or even opaque. While **TransDecomp** may struggle to extract ideal foreground pixels under these conditions, it compensates by generating a lower **Alpha value**, signaling to the pipeline that the surface is substantially non-transparent. This signal is vital for the subsequent **GeoSemTransNet** to prioritize surface-specific texture synthesis over background transmission.
>
> **Analysis of Failure Cases for Alpha Blending:**
>
> - **Object Thickness ($d$):** Lateral displacement $\Delta x$ increases linearly with thickness. Following geometric optics:$$\Delta x = d \cdot \frac{\sin(\theta_i - \theta_t)}{\cos(\theta_t)}$$
>     If $\Delta x$ exceeds a single pixel width or the human perception threshold for background continuity, the blending approximation breaks down.
>
> - **Surface Curvature:** High-curvature surfaces (e.g., lenses, spheres) act as optical elements, magnifying or flipping the background. When the focal length $f$ approaches the viewing distance, the resulting distortion becomes too extreme for linear Alpha Blending to simulate.
> - **Refractive Index Difference ($\Delta n$):** A larger $\Delta n$ results in a greater deflection angle and lower inherent transparency.
>
> **Summary:** While Alpha Blending has limitations, it primarily fails when the object's appearance approaches an opaque state that effectively diverges from the core problem our work aims to solve—the perception of **transparent objects**.. Furthermore, our training on the **ClearPose** dataset includes challenging scenarios (for instance, regions where the accumulation of overlapping transparent objects **results in an opaque appearance**) where weak refraction is violated. In these cases, **GeoSemTransNet** leverages its generative capability to synthesize reasonable surface pixels, even when foreground cues are imperfect.
>
> ---
>
> ### **Q2. Explanation for Performance Drop in Ablation Studies**
>
> To be candid, our exploration and progress in these experiments were essentially the inverse of the ablation process—a fact you have astutely recognized regarding the extensive efforts behind our work.
>
> We initially aimed to propose a fully end-to-end approach for re-editing transparent regions; however, after extensive verification, we found that without sufficiently reliable signals and geometric priors, it is difficult for a model to modify 2D pixel values to meet the requirements of 3D tasks. A primary reason is the current lack of a sufficiently reliable depth estimation or reconstruction model that can act as a "proxy" to predict depth from our synthesized images during training. Consequently, we could not directly employ 3D-related loss functions or training strategies to supervise the 2D editing process.
>
> Therefore, we integrated modules capable of predicting reliable priors to facilitate the final pixel modification. Since these intermediate modules were designed specifically to serve the final **GeoSemTransNet**, an incomplete preprocessing pipeline reverts to a dilemma of lacking reliable priors. This is why the performance of partial configurations in our ablation studies is inferior to that of original images without any preprocessing.

---

> > ### Author Rebuttal · Reviewer_8oR8 · 2026-04-04
> >
> > Thanks for the rebuttal. The additional explanations fully address my concerns. I will keep my score.

---

### Decision · Program_Chairs · 2026-04-30

**Decision:**

Accept (regular)

**Comment:**

This paper presents GHOST, a preprocessing framework that transforms transparent objects into opaque RGB representations, allowing off-the-shelf 3D vision models to process them accurately without retraining.

Reviewers praised the architecture and real-time efficiency. The authors' rebuttal resolved initial concerns regarding the "weak refraction" assumption, dataset generalization, and resolution settings by providing zero-shot TROS evaluations and clarifying optical limits. This secured a unanimous positive consensus.

Therefore, the AC recommends Acceptance.

Camera-Ready Note: Please ensure the new zero-shot TROS evaluations, in-the-wild visualizations, and the discussion on the limitations of the "weak refraction" assumption are fully integrated into the final manuscript.